# Control Structure Design Using Global Sensitivity Analysis for Mineral Processes under Uncertainties

Oscar Mamani-Quiñonez [1], Luis A. Cisternas [1], Teresa Lopez-Arenas [2] and Freddy A. Lucay [3,*]

[1] Departamento de Ingeniería Química y Procesos de Minerales, Universidad de Antofagasta, Antofagasta 1240000, Chile; oscar.maquiz@gmail.com (O.M.-Q.); luis.cisternas@uantof.cl (L.A.C.)
[2] Departamento de Procesos y Tecnología, Universidad Autónoma Metropolitana-Cuajimalpa, Ciudad de México 05348, Mexico; mtlopez@cua.uam.mx
[3] Escuela de Ingeniería Química, Pontificia Universidad Católica de Valparaíso, Valparaíso 2340000, Chile
*   Correspondence: freddy.lucay@pucv.cl

**Abstract:** Multiple-input and multiple-output (MIMO) systems can be found in many industrial processes, including mining processes. In practice, these systems are difficult to control due to the interactions of their input variables and the inherent uncertainty of industrial processes. Depending on the interactions in the MIMO process, different control strategies can be implemented to achieve the desired performance. Among these strategies is the use of a decentralized structure that considers several subsystems and for which a SISO controller can be designed. In this study, a methodology based on global sensitivity analysis (GSA) to design decentralized control structures for industrial processes under uncertainty is presented. GSA has not yet been applied for this purpose in process control; it allows us to understand the dynamic behavior of systems under uncertainty in a broad value range, unlike approaches proposed in the literature. The proposed GSA is based on the Sobol method, which provides sensitivity indices used as interaction measures to establish the input–output pairing for MIMO systems. Two case studies based on a semi-autogenous grinding (SAG) mill and a solvent extraction (SX) plant are presented to demonstrate the applicability of the proposed methodology. The results indicate that the methodology allows the design of $2 \times 2$ and $3 \times 3$ decentralized control structures for the SAG mill and SX plant, respectively, which exhibit good performance compared to MPC. For example, for the SAG mill, the determined pairings were fresh ore flux/fraction of mill filling and power consumption/percentage of critical speed.

**Keywords:** global sensitivity analysis; uncertainty; control structure; SX process; SAG mill

## 1. Introduction

Most industrial control systems are multiple-input and multiple-output (MIMO) systems, as the goal of multivariable control includes keeping multiple variables controlled at independent set points. For instance, mining plants, oil refineries, biorefineries, and, in general, chemical manufacturing plants contain MIMO processes. In these systems, each manipulated variable (input) can affect several controlled variables (outputs) causing interactions between them and consequently generating coupling in the system. In practice, such interactions result in difficulties in analyzing and controlling a given system. Furthermore, the parameters used to define the input and output variables may present uncertainty, vary with time, or be unknown [1]. For these reasons, the analysis of how to control MIMO systems is often more complex compared to single-input and single-output (SISO) systems.

Depending on the interactions in the MIMO process, different control strategies can be applied to achieve the desired performance: decentralized, centralized, or decoupled control. The centralized structure considers the design of a complete multivariate controller to control n output variables using $n$ manipulated variables, yielding that $n^2$ number of controllers prevail. However, these control systems are complex and lack integrity [2].

The decentralized structure considers several subsystems for which a SISO controller is designed. Thus, only $n$ controllers prevail for each $n$ output variable, since it uses single-loop or diagonal controllers [3]. The decoupling structure uses separate elements, known as decouplers, or simply controllers, to compensate for the strong interactions present in the system [4]. Decoupling can be divided into static and dynamic decoupling according to the characteristics of time, or can be classified into total and approximate decoupling according to the degree of decoupling.

The decentralized control technique is still widely used in many industrial control systems due to its simple implementation, proficient maintenance, simple tuning, and robust performance even under model mismatches and uncertainties [5]. The key issue when designing a decentralized control system is the control structure design (CSD), that is, the selection of inputs and outputs and how they are paired [6]. The available literature proposes several mathematical measures to quantify the degree of interaction between input–output pairs. Probably the most widely used measure is relative gain array (RGA), which was proposed by Bristol in 1966 and requires only the steady-state gain of the plant model. This information can be obtained by step test methods. The simplicity of RGA is the main reason for its popularity [7]. RGA has been studied and used by several authors to propose new interaction measures. For example, Niederlinski [8] proposed the use of an index based on the gain matrix to provide direct information on the ability of a decentralized control to stabilize a $2 \times 2$ MIMO system. A variation of RGA was reported by Zhu [9], known as relative interaction array (RIA). This is based on the concept of viewing the interaction as an unmodeled term for a particular pairing. A dynamic extension of RGA was proposed by Kinnaert [10] which can be applied to analyze plants at any frequency. Mc Avoy et al. [11] also proposed a dynamic extension of RGA which assumes the availability of a dynamic process model that is used to design an optimal proportional output controller.

RGA provides limited knowledge; specifically, it does not indicate when to use multivariable controllers or how to carry out CSD. Therefore, some authors have proposed alternative approaches. Salgado and Conley [12] considered observability and controllability Gramians in so-called participation matrices (PMs). Using a similar approach, Wittenmark and Salgado [13] introduced the Hankel interaction index matrix (HIIA). These Gramian-based interaction measures help to overcome most of the disadvantages of RGA. Specifically, these measures seem to provide suggestions for designing controller structures. Hanzon [14] showed that the PM is closely related to the direct Nyquist array, which was introduced by Rosenbrock in 1970. Birk and Medvedev [15] proposed an alternative to HIIA. They used the $\mathcal{H}_2$ and $\mathcal{H}_\infty$ norms as the basis for new interaction measures. Meanwhile, Halvarsson et al. [16] proposed a different approach to obtain interaction measures based on linear quadratic Gaussian (LQG) control. Moreover, many MIMO systems present uncertainty, creating a set of possible systems for which the interaction measures may differ. Consequently, the control structure design (CSD) may differ between models. For example, Jain and Babu [17] analyzed the sensitivity of RGA to model uncertainty. Specifically, they studied how the process dynamics can affect CSD decisions proposed by RGA in systems under uncertainty.

As outlined above, there are currently many rigorous methods of CSD based on process control theory. On the other hand, there is a large gap between research and industrial application, which means process control engineers in industry today still use a strongly empirical approach to CSD, basing their decisions on practical knowledge or principles of common sense and experience [6]. For instance, in the milling process, input–output pairing has been established based on practical knowledge or trial and error under uncertainty [18–20], or not [21–23]. In solvent extraction (SX) [24], froth flotation [25], and melting furnaces [26], input–output pairing was settled using classical RGA. These works verify the need to have one methodology to help establish CDS under uncertainty for devices implemented in mineral processing.

Within this context, global sensitivity analysis (GSA) is proposed in this work as an alternative to decide on the CSD. Sensitivity analysis (SA) is a commonly used method of identifying the important input variables that determine the behavior of a model under uncertain conditions. SA can be performed locally or globally, and according to Saltelli [27], the latter is more robust and reliable even for nonlinear models. There are several methods of performing GSA and among them, due to their versatility and efficiency, those based on variance decomposition stand out [28]. Because these methods involve high computational cost, Homma and Saltelli [29] introduced the concept of total sensitivity indices to overcome this disadvantage. These indices indicate the average effect of a given input variable on a specific output of the model, taking into account all possible interactions with the other input variables of the system. A significant advantage of this method is that it can be used in both steady-state and dynamic systems [27].

In this work, GSA is proposed to determine the CSD for nonlinear MIMO systems under time-varying and uncertain conditions. GSA has not yet been applied to this purpose in process control; it allows us to understand the dynamic behavior of systems under uncertainty in a broad value range, unlike approaches proposed in the related literature. The first order and total sensitivity indices provided by GSA are used as measures of input–output interactions for the CSD. The methodology is illustrated with two case studies from the mining industry: a semi-autogenous grinding (SAG) mill and an SX plant. The open-loop modeling and simulation of these processes have been studied previously [30,31], so in this work, the analysis of CSD based on GSA is presented first. Subsequently, for the purpose of comparing control performance levels, several controllers were designed based on different control structures (reported structures vs. CSD proposed in this work) and control strategies (proportional–integral (PI) control and model predictive control (MPC)). The results of the GSA in both cases studies were obtained using the Sobol–Jansen method, which allowed quantification of the interactions of variables over time as well as observation of the changes in the output variables due to the uncertainty of input variables. The information generated by GSA allowed a reduction of the CSD of the SAG process from a $3 \times 3$ to a $2 \times 2$ MIMO system, and that of the SX process from a $4 \times 5$ to a $3 \times 3$ MIMO system. Finally, these control structures were implemented for the control strategies described above and better closed-loop performance was obtained when reduced CSDs were implemented.

## 2. Materials and Methods

### 2.1. Uncertainty Analysis (UA)

Mathematical models are fundamental tools in decision-making and are developed considering assumptions and sometimes little-known information, introducing uncertainty in the modeling. Uncertainty can be classified as either stochastic or epistemic [32]. The former is also known as variability, inherent uncertainty, irreducible uncertainty, or uncertainty due to chance and is related to variations inherent in a given system, usually as a result of the random nature of model inputs. The latter is also known as reducible uncertainty, subjective uncertainty, or uncertainty due to a lack of knowledge. This uncertainty type, as a source of non-deterministic behavior, derives from a lack of knowledge of the system or the environment. Uncertainty in numerical models has many origins: input data, model simplification, algorithm structure, calibration process, calibration and validation data, and equifinality. In this context, UA corresponds to determining the uncertainty in the output variables as a result of the uncertainty in the input variables. UA can be addressed using probability theory, imprecise probability, probability bound analysis, evidence theory, or possibility theory [33]. In this work, UA is applied using probability theory, with a procedure that includes four steps: first, the uncertain input variables are described using the probability distribution function (PDF); second, a sample is generated from the PDF using random sampling, such as the Monte Carlo method; third, the values of the model output variables are determined for each element of the sample; fourth, the behavior of the model output variables is characterized by graphs, descriptive statistics, and statistical tests.

### 2.2. Sensitivity Analysis (SA)

According to Saltelli [34], SA can be defined as an examination of how the uncertainty in the output of a model can be apportioned among different sources of uncertainty in the model's input variables. This analysis can be done locally or globally. The latter quantifies the importance of model inputs and their interactions with respect to model outputs. GSA provides an overall view of the influence of inputs on outputs, as opposed to the local view based on partial derivatives, which has the disadvantage of depending on the choice of the evaluation point. The general objectives of GSA are as follows [35]: to identify significant and insignificant variables in a given model, aiming to reduce its dimension; to improve the understanding of model behavior, specifically highlighting interactions between input variables and finding combinations of input variables that result in high or low values for the model output. GSA considers six steps [36]: (1) determine the objective function, (2) select the input variables of the model, (3) assign a range and type of PDF to the input variables, (4) apply a sampling design to generate samples, (5) assess the model for the generated samples, and (6) implement the results of step 5 to perform GSA and determine the importance of the input variables on the model outputs. The related literature reveals that there are several methods of performing GSA and those based on variance decomposition are used more often due to their versatility and efficiency [28]. In this category, approaches based on the method of Sobol can be found. The latter considers a squared-integrable function $f$ on $\Omega^m = \{x/0 \leq x_j \leq 1, \ j = 1, 2, \ldots, m\}$ that is represented in terms of increasing dimensions [27]:

$$f = f_0 + \sum_j f_j + \sum_j \sum_{k>j} f_{jk} + \ldots + f_{1,2,\ldots,m} \tag{1}$$

where $f_j = f_j(x_j)$, $f_{jk} = f_{jk}(x_j, x_k)$, and so on; whereas $f_0 = E(Y)$, $f_j = E(Y/x_j)$, $f_{jk} = E(Y/x_j, x_k) - f_j - f_k - E(Y)$, and so on. Here, $Y = f(x_1, x_2, \ldots, x_m)$ and $E$ represents the mathematical expectation. Note that these last expressions have the following properties: $V_j = V(f_j(x_j)) = V(E(Y/x_j))$, $V_{jk} = V\left(f_{jk}(x_j, x_k)\right) = V(E(Y/x_j, x_k)) - V(E(Y/x_j)) - V(E(Y/x_k))$, and so on. Here, $V$ represents the variance. The square integration of Equation (1) on $\Omega^m$ allows us to obtain the so-called ANOVA-HDMR decomposition or its normalized equivalent:

$$V(Y) = \sum_j V_j + \sum_j \sum_{k>j} V_{jk} + \ldots + V_{1,2,\ldots,m} \tag{2}$$

$$1 = \sum_j \frac{V_j}{V(Y)} + \sum_j \sum_{k>j} \frac{V_{jk}}{V(Y)} + \ldots + \frac{V_{1,2,\ldots,m}}{V(Y)} \tag{3}$$

In Equation (3), $j = 1, 2, \ldots, m$, $V(Y)$ represents the model variance, $V_j$ represents the first order effect for each input variable $x_j$, and $V_{jk}$ to $V_{1,2,\ldots,m}$ represent the interactions of the $m$ input variables. The calculation of Equation (3) has a high computational cost that can be overcome by calculating total sensitivity indices [29]. These indices allow us to determine the average effect of a given input variable, considering all possible interactions of the respective variable with all other input variables. In this work, the Sobol–Jansen method was used, which allows calculation of the first order sensitivity index ($S_j$) and the total sensitivity index ($S_j^T$) for input variable $x_j$ of the mathematical model. The Sobol–Jansen method has been used to analyze flotation circuits [37,38], heap leaching [31], grinding [30], and the lithium supply chain [39]. In addition, it exhibits high performance when analyzing chemical processes [40]. This method considers 5 steps [28]: first, choose an integer $N$; second, generate a matrix of size $(N, 2r)$ of quasi-random numbers from the sampling of input variables of their respective PDF ($r$ represents the number of input variables); third, divide the matrix into 2 submatrices, $A$ and $B$, of size $(N, k)$; fourth, form matrix $D_j$ from the columns of matrix $A$, except the $j$th column, which is taken from matrix $B$, and similarly,

form matrix $C_j$ from the columns of matrix $B$, except the $j$th column, which is taken from matrix $A$; fifth, assess the model output in matrices $A$, $B$, $C_j$, and $D_j$, obtaining $Y_A = f(A)$, $Y_B = f(B)$, $Y_{C_i} = f(C_i)$, and $Y_{D_i} = f(D_i)$, and subsequently use the following equations:

$$S_j = \frac{V\left(E\left(Y/x_j\right)\right)}{V(Y)} = \frac{V(Y) - \frac{1}{2N}\sum_{i=1}^{N}\left(Y_B^{(i)} - Y_{D_j}^{(i)}\right)^2}{V(Y)}, j = 1, 2, \ldots, m \tag{4}$$

and

$$S_j^T = 1 - \frac{V\left(E\left(Y/x_j\right)\right)}{V(Y)} = \frac{\frac{1}{2N}\sum_{i=1}^{N}\left(Y_A^{(i)} - Y_{D_j}^{(i)}\right)^2}{V(Y)}, j = 1, 2, \ldots, m \tag{5}$$

Other expressions to estimate the sensitivity indices can be found elsewhere [28,36,40]; these use one or another matrix defined earlier, e.g., the Sobol–Jansen method uses matrices $A$, $B$, and $D$. The interpretation of the indices is straightforward: the higher the sensitivity index of an input variable, the greater its influence on the model output. The first order index allows us to determine the most important input variable, while the total sensitivity index allows us to identify the input variables that do not influence the model outputs. In this sense, if input variable $x_j$ of the model does not interact with the other input variables, the sensitivity indices satisfy $S_j \approx S_j^T$, otherwise $S_j < S_j^T$. If $S_j^T \approx 0$, input variable $x_j$ does not influence the model output and can be fixed at its nominal operating value and consequently the dimension of the mathematical model can be reduced [27]. Note that, ideally, UA precedes SA, as before uncertainty can be apportioned, it needs to be estimated [41].

### 2.3. Solving the Model in MATLAB–Simulink

The models were implemented as a Mask subsystem in MATLAB–Simulink[TM]. Simulink[TM] (R2020a-Academic Version) is a programming system that uses blocks, i.e., graphical programming, to solve differential equations. In this work, such equations were solved using the ode4 solver based on the fourth order Runge–Kutta formula. In addition, Simulink[TM] allows users to program their own blocks across functions. This feature and the possibility to use specific toolboxes, such as PID control and MPC, provide a powerful platform for the development of prototypes.

### 2.4. Methodology for Control Structure Design (CSD)

A computational method of developing control structures is proposed and presented in Figure 1. In the first step, the multivariable system is modeled using mathematical and computational tools, such as differential equations and MATLAB software (R2020a-Academic Version), respectively. In the second step, the process variables are classified as manipulated, controlled, supervised, or disturbed. Furthermore, the variables manipulated under uncertainty are characterized by distribution functions after determining the type of uncertainty. In the third step, GSA is carried out using methods based on variance decomposition, such as the Sobol–Jansen method, after carrying out UA. Here, the GSUA toolbox [42] is implemented, and the sample size for each uncertain manipulated variable is defined as one thousand. According to [43], this value allows us to obtain robust results from UA and GSA. Subsequently, the input–output pairing is selected according to the total sensitivity indices provided by GSA. Here, it was important to analyze the behavior of total sensitivity indices over the simulation time to establish such pairing, which allowed us to obtain a decentralized structure whose SISO subsystems can be controlled using PID or PI controls. In the fourth step, the designed control structure is evaluated through simulations and experiments. Specifically, the control structure is subject to different set points and compared with other approaches proposed in the related literature. If the control structure provides satisfactory results, it is considered robust; otherwise, we return to the second step. The latter considers changing the nominal operating conditions or parameters used to define the distribution functions.

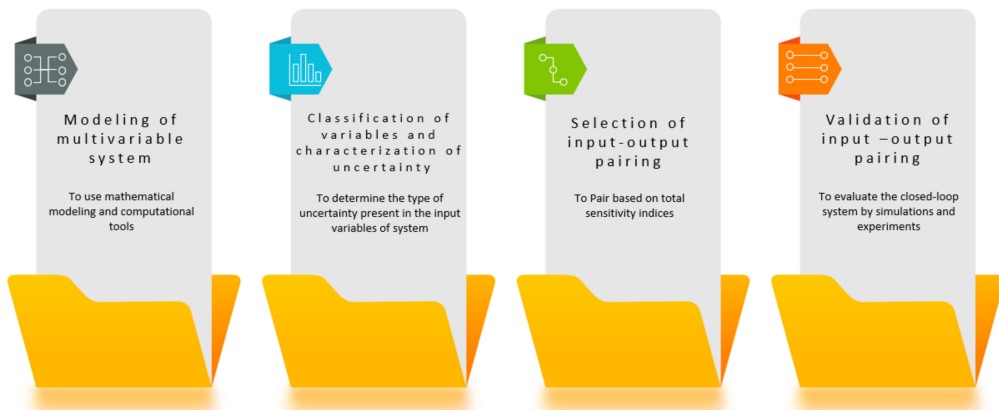

**Figure 1.** Methodology used for CSD using GSA and UA.

## 3. Results

The methodology proposed for CSD is illustrated considering grinding and SX processes to demonstrate the methodology's capacity to address systems with different degrees of freedom.

### 3.1. Semi-Autogenous Grinding (SAG)

Step 1. Modeling

From an energy point of view, mineral milling is decisive in the evaluation of operating cost, representing 50–80% of the total operating cost of a mineral concentrator plant. Various modeling trends are proposed in the literature based on the principles that govern the grinding phenomenon; this is how the models based on population models stand out, which were used for this case study. The milling model presented by Austin et al. [44,45] was considered. The SAG mill model is generally divided into two zones, the grinding chamber and the sorting zone, as shown in Figure 2. The $F$ particles entering the mill are introduced into the grinding chamber. The product obtained, $P^*$, faces the classification zone, where, according to a classification probability $c_i$, the particles can return to the crushing chamber or become part of product $P$ of the SAG mill.

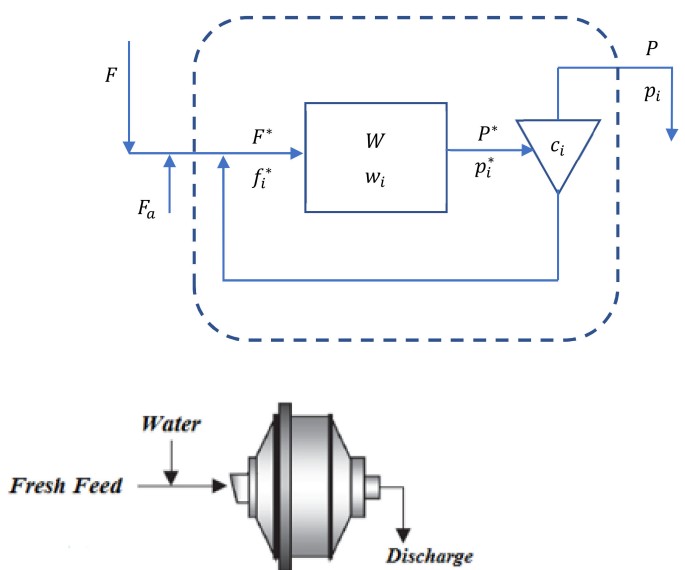

**Figure 2.** Schematic representation of a SAG mill, adapted from [30].

It is commonly assumed that the SAG mill behaves like a perfectly mixed reactor, with a mass retained ($W$) in the volume ($V$) of the mill and first order kinetics. Some

authors indicate that second order kinetics might be able to model the breakage of coarse particles better than first order kinetics [46,47]. However, computational experiments were carried out under conditions where first order kinetics provided reliable estimations [48,49]. According to Austin et al. [44], $F$ is the fresh ore flux fed to mill and the $i$th size fraction in $F$ is $K_i w_i W$. In this expression, $w_i$ is the weight fraction of retained mass in the mill and $K_i$ is the specific breakage rate of the $i$th size fraction. When a fraction of size $i$ breaks, a fraction $b_{ij}$ of the broken material is sent to size $j$. The dynamic mass balance in each size $i$ is:

$$\frac{d[w_i(t) \cdot W]}{dt} = F_i - P_i + W \sum_{\substack{j=1 \\ i>j}}^{i-1} b_{ij} K_j w_j - K_i w_i W,$$

$$w_i(0) = w_{i0}, \ i = 1, \ 2, \ \ldots, n; n \geq i \geq j \geq 1 \tag{6}$$

where $n$ is the number of species present in the fresh feed, $F_i$ is the fraction of ore flux ($F$) fed to the mill, $P_i$ is the fraction of flux discharged ($P$), and $c_i$ is the classification efficiency of the internal grid, which affects the mass flow recirculated internally ($C^* = \sum_i c_i w_i / \sum_i w_i (1 - c_i)$). The complete model equations and parameter values can be found in Appendix A and [30]. In the appendix, the reader can see the expressions used to model the cumulative breakage distribution function and its implementation to determine $b_{ij}$, as well as expressions used to estimate the classification efficiency of the internal grid mill, which is required to calculate recirculated mass flow. In [30], the reader can find comminution-specific energy, mill power consumption, and the fraction of mill filling expressions, among other equations.

The design of the grinding process by Magne et al. [49] is considered in this work. Here, a SAG mill was implemented to process copper sulfide ore 1.83 m in diameter ($D$) and 0.61 m in length ($L$). The operating conditions of the SAG mill were as follows: ore flux fed ($F$) at 3.45 t/h with granulometry of 12% for 4″, 8% between 4″ and 2″, and 80% below 2″; mill volume occupied by the discharge mill ($J_b$) equal to 8.5% by volume; percentage of solids in the discharge mill ($Y_d$) equal to 74%; operating speed equal to 72% of critical speed ($\varphi_c$); flow of water fed ($F_a$) equal to 1.2 m$^3$/h; a classification grill with an opening of $\frac{1}{2}$″. The SAG mill model was simulated using Simulink, obtaining fraction of mill filling ($J$), power consumption ($M_p$), and retained mass ($W$) equal to 0.22, 9.8 kW, and 0.41 t/h, respectively.

Step 2. UA and GSA

For the MIMO system (see Figure 3a), the main output variables are $J$, $M_p$, and $W$, while the possible manipulated input variables are $F$, $F_a$, $\varphi_c$, $F_1, F_2$, and $F_3$. The manipulated input variables were described using distribution functions that allow the effect of uncertainty in the system to be included. In this context, the SAG mill feed fractions exhibit stochastic uncertainty due to geological uncertainty, while the other SAG mill input variables exhibit epistemic uncertainty due to insufficient measurements, as reported in [49]. Considering that the particle size distribution of the feed to the mill can be represented by a normal distribution [50] and that the particle fragmentation exhibits a fractal nature [51], the fractions in the SAG mill feed are described using the normal distribution. According to the principle of indifference, a uniform distribution should be implemented to describe epistemic uncertainty in the absence of information [52]. Then, manipulated input variables were described as follows: $F \sim U[3.24, 3.65]$ t/h, $F_a \sim U[1.02, 1.39]$ m$^3$/h, $\varphi_c \sim U[0.7, 0.74]$, $F_1 \sim N[12, 0.70]$%, $F_2 \sim N[8, 0.80]$%, and $F_3 \sim N[80, 0.73]$%.

The MIMO system for the SAG mill is shown in Figure 3a, and Figure 3b–d show the UA results considering a sample of 6000 instances of operation. Here, it can be observed that $J$, $M_p$, and $W$ present values around the responses obtained using nominal operating conditions (red line) when the SAG mill was simulated under operational uncertainty, which is consistent with previously reported results [30,49]. Thus, the grinding model provides robust estimates under uncertainty; subsequently, the uncertainty must be apportioned. The GSA results using the Sobol–Jansen method are shown in Figure 4.

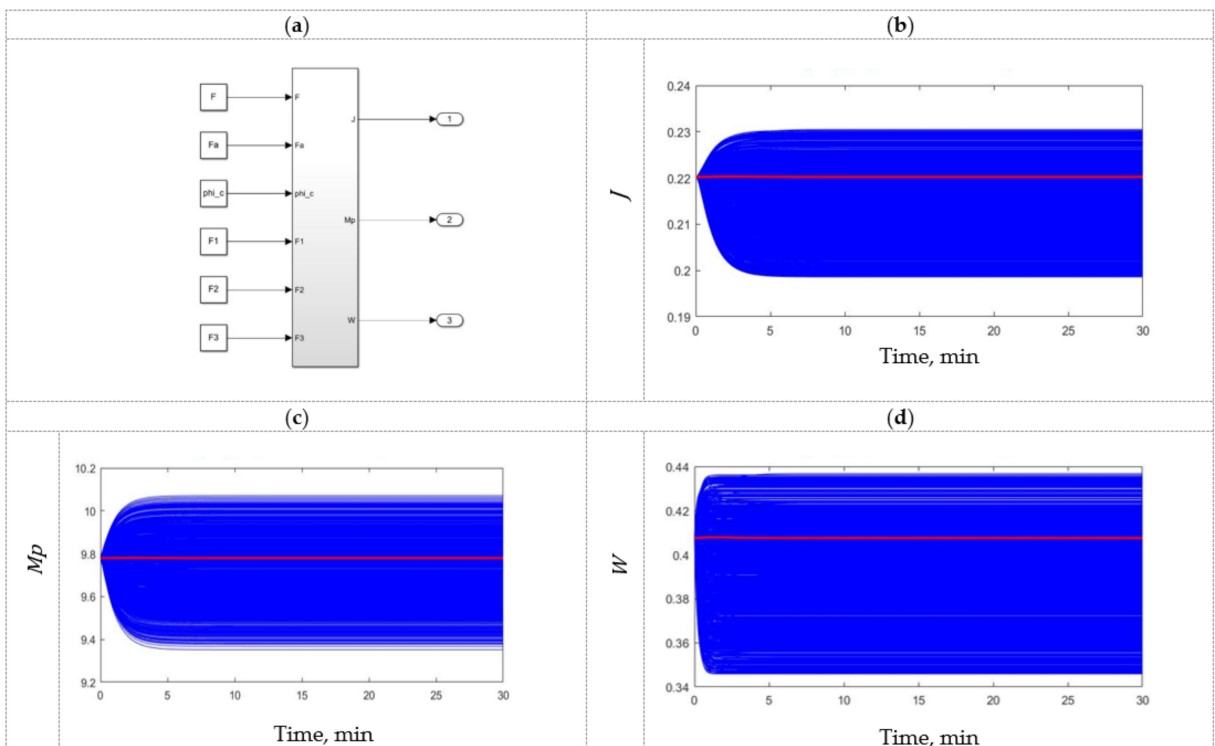

**Figure 3.** (**a**) SAG mill as MIMO system, and UA results using (**b**) mill filling, (**c**) power consumption, and (**d**) mass retained as output variables.

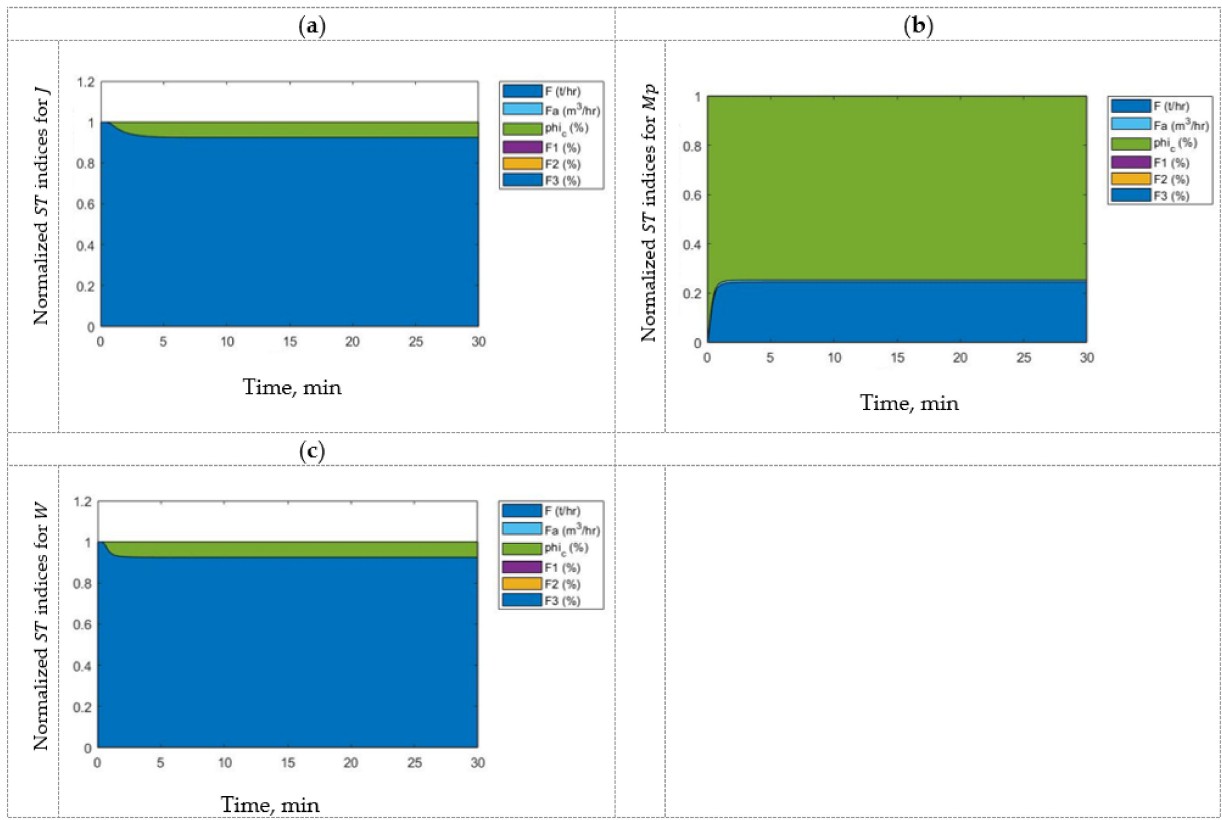

**Figure 4.** GSA results using (**a**) mill filling, (**b**) power consumption, and (**c**) mass retained as output variables.

Figure 4 shows the normalized total sensitivity indices of the input variables for each output variable of the SAG mill. Here, the first-order indices are not shown because the total sensitivity indices provide more relevant information for the purpose of this work. According to these results, it can be concluded that all output variables $(J, M_p, W)$ strongly depend on $F$ and $\varphi_c$, and there is a negligible effect of $F_a$, $F_1$, $F_2$, and $F_3$. Then, the input–output pairing is selected using this information. These results suggest that the manipulated input variables should be $F$ and $\varphi_c$, while the disturbances should be $F_a$, $F_1$, $F_2$, and $F_3$.

Step 3. CSD

Now, the process variables can be classified as manipulated, controlled, supervised, or disturbed, allowing the SAG mill model to be expressed in a standard control notation (Figure 5) as follows:

$$\dot{x} = f(x, u, d, \ p), \ x(0) = x_0 \ ; \ \dim(x) = 3 \ , \ \dim(u) = n_u \tag{7}$$

$$y = h(x) \ ; \ dim(y) = n_y \tag{8}$$

where the vector-valued function of time $f$, defined by the right-hand side of Equation (6), depends on vectors of states $(x)$, manipulated input variables $(u)$, disturbances $(d)$, and model parameters $(p)$. For the SAG mill, the states $(x)$ correspond to the weight fraction of the mill retention $(w_i)$, while the outputs $(y)$ in Equation (8) are the variables to be regulated at desired values (controlled variables).

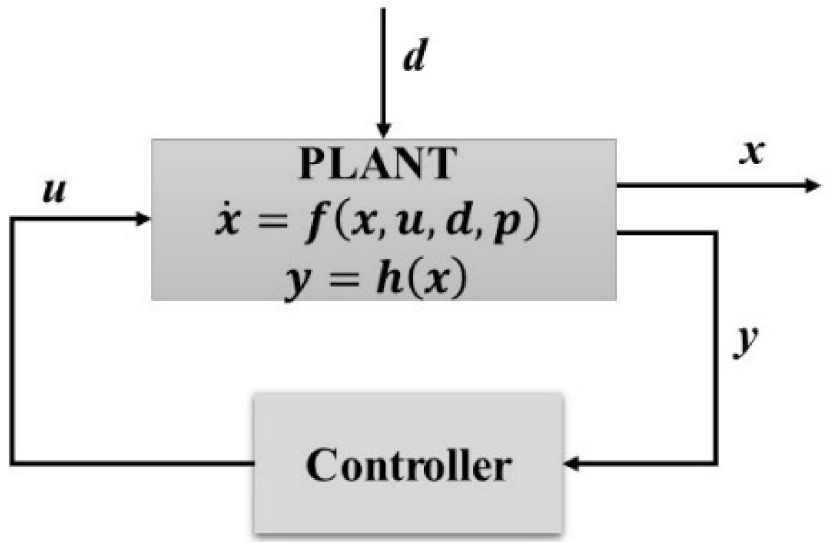

**Figure 5.** General control system.

For the MIMO system of the SAG mill (see Figure 3a), and for comparison purposes, two structures are defined for the CSD:

(a)  A traditional structure (3 × 3) previously reported for the SAG mill [53], with $n_u = n_y = 3$:

$$u = \begin{bmatrix} F \\ F_a \\ \varphi_c \end{bmatrix} , \quad y = \begin{bmatrix} J \\ M_p \\ W \end{bmatrix} , \quad d = \begin{bmatrix} F_1 \\ F_2 \\ F_3 \end{bmatrix} , \quad x = \begin{bmatrix} w_1 \\ w_2 \\ w_3 \end{bmatrix} \tag{9}$$

(b) A reduced structure (2 × 2) obtained in step 2 (considering $W$ as a supervised variable, which is kept in a range depending on grinding design capacity [54]), with $n_u = n_y = 2$:

$$u = \begin{bmatrix} F \\ \varphi_c \end{bmatrix}, \quad y = \begin{bmatrix} J \\ M_p \end{bmatrix}, \quad d = \begin{bmatrix} F_1 \\ F_2 \\ F_3 \\ F_a \end{bmatrix}, \quad x = \begin{bmatrix} w_1 \\ w_2 \\ w_3 \end{bmatrix} \tag{10}$$

In addition to the two control structures designed using total sensitivity indices, two other control strategies are considered to evaluate performance and robustness: a conventional proportional–integral–derivative (PID) controller and an advanced MPC. So, in the next step, three control schemes are evaluated and analyzed, as shown in Figure 6: 2 × 2 PID control, 2 × 2 MPC, and 3 × 3 MPC.

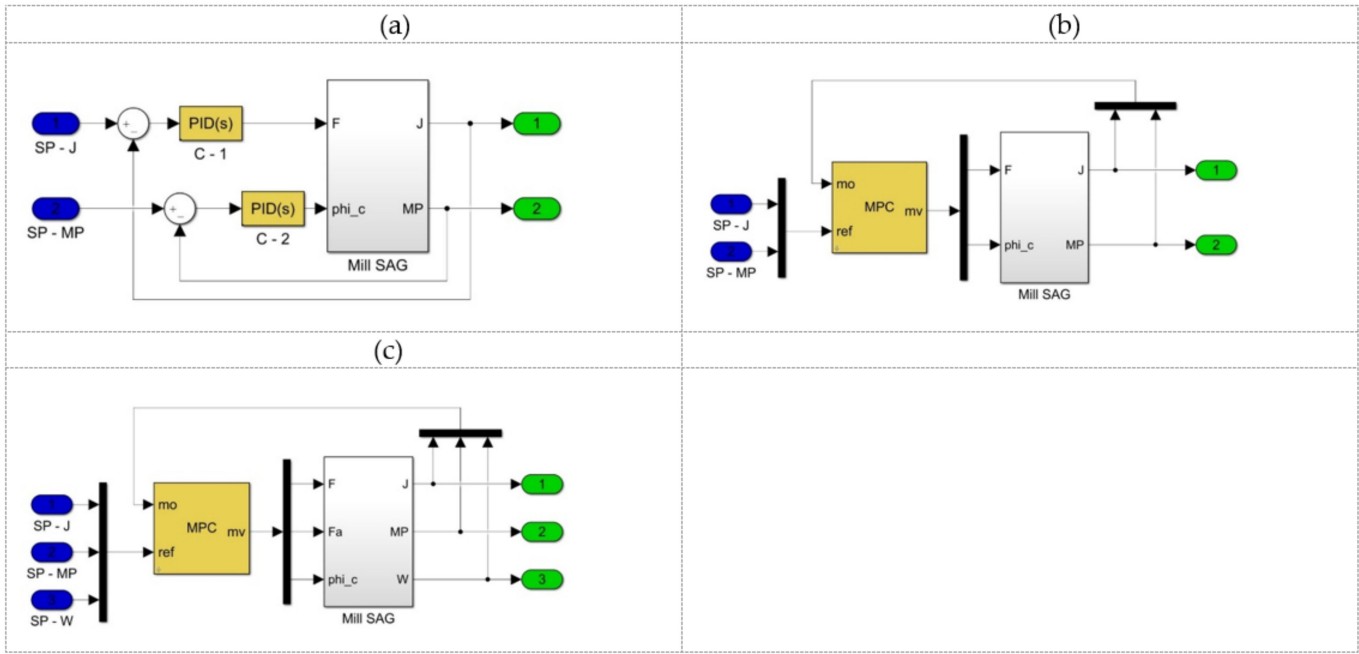

**Figure 6.** Control systems for SAG mill: (**a**) 2 × 2 PID control, (**b**) 2 × 2 MPC, (**c**) 3 × 3 MPC.

Step 4. Closed-loop validation

Subsequently, the designed control systems were implemented in MATLAB and compared with their corresponding open-loop dynamics, as shown in Figure 7. It is important to note that the tuning parameters for the PID and MPC controllers were determined using the automatic tuning tool included in Simulink. It can be seen in Figure 7a,b that the SAG mill load and energy consumption responses for a step change in the set point are satisfactory. To quantify the quality of these results, the integral absolute error (IAE) was calculated using the formula $IAE = \int |e(t)| dt$, where $e(t)$ is the difference between the set point and the controller response [55]. Table 1 shows a summary of IAE values obtained for the three controllers.

In the case of $J$, Table 1 shows that the 2 × 2 PID, 3 × 3 MPC, and 2 × 2 MPC controllers provide IAE values of 0.001, 0.020, and 0.075, respectively. Thus, the PID control designed using the pairing $J/F$ proposed by the CSD methodology provides high performance compared to the other controllers. In the case of $M_p$, Table 1 shows that the 3 × 3 MPC, 2 × 2 MPC, and 2 × 2 PID controllers provide IAE values of 0.111, 0.299, and 0.630, respectively. Therefore, the PID control designed using the pairing $M_p/\phi_c$ proposed by the CSD methodology demonstrates sufficient performance compared to the others. In the first case, the high performance might be based on the strong influence of the feed flux on

mill load, regardless of time and uncertainty (approximately 0.95; see Figure 4a). In the second case, the percentage of critical velocity and feed flux influence power consumption 0.75/0.25 regardless of time and uncertainty, this feed flux effect could explain the sufficient performance of the $2 \times 2$ PID controller. This could also explain the offset observed in the behavior of manipulated variables (Figure 7d).

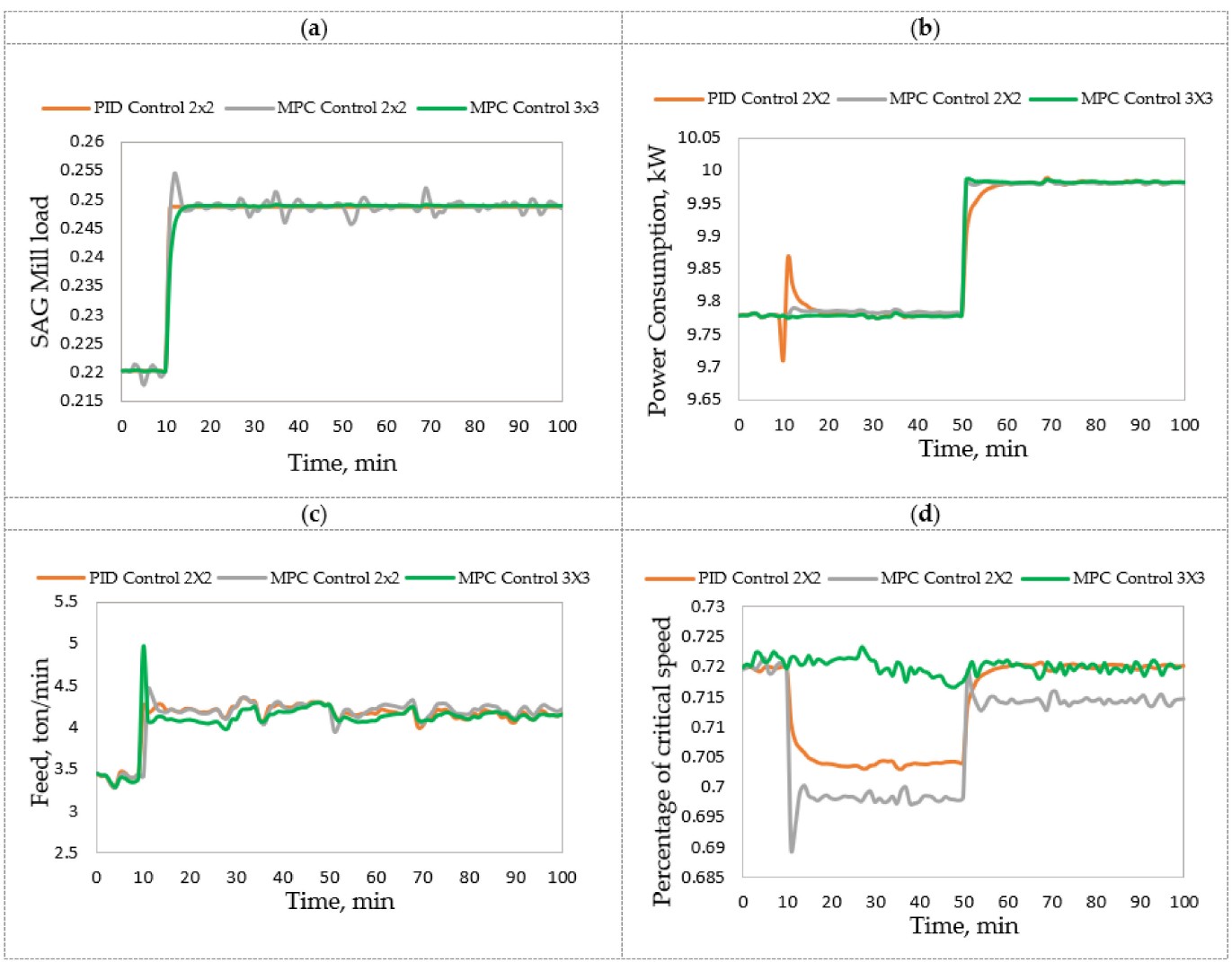

**Figure 7.** Dynamic responses of controlled variables: (**a**) SAG mill load ($J$); (**b**) power consumption ($M_p$) and manipulated variables; (**c**) feed ($F$); (**d**) percentage of critical speed ($\varphi_c$).

**Table 1.** Analysis and interpretation of results obtained by controllers, SAG mill.

| Control | | Controlled Variables | | Manipulated Variables | |
|---|---|---|---|---|---|
| | IAE | $J$ | $M_p$ | $F$ | $\phi_c$ |
| High performance | | PID $2 \times 2$ 0.001 | MPC $3 \times 3$ 0.111 | PID $2 \times 2$ 6.437 | PID $2 \times 2$ 0.793 |
| Good performance | | MPC $3 \times 3$ 0.020 | MPC $2 \times 2$ 0.299 | MPC $3 \times 3$ 6.081 | MPC $3 \times 3$ 0.240 |
| Sufficient performance | | MPC $2 \times 2$ 0.075 | PID $2 \times 2$ 0.630 | MPC $2 \times 2$ 9.594 | MPC $2 \times 2$ 0.995 |
| Reference | | Open-loop | | | |
| | | 0.255 | 11.536 | | |

### 3.2. Solvent Extraction (SX) Process

Step 1. Modeling

The SX process includes an extraction and a re-extraction system (Figure 8), which can be located in different configurations, both in parallel and in series. The model proposed by Komulainen et al. [56] consists of four units, three for extraction and one for re-extraction. The flow input to the process comprises pregnant leach solutions (PLS) with mineral to be recovered, $F_{1a}$ and $F_{2a}$; solution with a copper concentration, $c_{0a}$; poor electrolyte flow, $F_{1e}$, with a concentration of $c_{0e}$; the flow of the loaded organic (LO) solution, $F_{LO}$. The LO solution is recycled in the process, but the flow can be managed through the organic storage tank, $c_{3o}$. The results of the process are rich copper concentration, $c_{1e}$, and refined copper concentrations, $c_{1a}$ and $c_{3a}$.

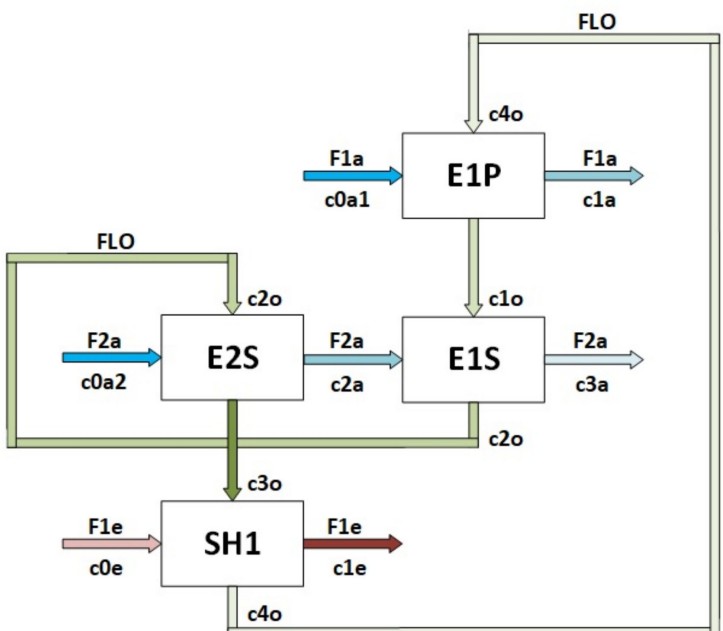

**Figure 8.** Schematic representation of SX process, adapted from [24].

In the extraction, the copper is transferred from the aqueous to the organic phase. Each of the three extraction units is modeled using dynamic mass balances for organic ($c_{io}$), aqueous ($c_{ia}$), and electrolyte ($c_{ie}$) phases:

$$\frac{dc_{i,phase}(t)}{dt} = \frac{F_{i,phase}(t)}{V_{mix,i}(t)}\left[c_{i-1,phase}(t-t_0) - c_{i,phase}(t)\right] + K_i\left[c_{i,phase}(t) - c^*_{i,phase}(t)\right]$$

$$c_{i,phase}(0) = c_{i,phase,0} \tag{11}$$

$$i = 1, \ldots, 4 \; for \; phase = o; \; i = 1,2,3 \; for \; phase = a; i = 1 \; for \; phase = e$$

where $c_i$ represents concentrations, $F_i$ represents flow rates, $V_{mix}$ represents the mixing volumes, $K_i$ represents the mass transfer coefficients, and the settler, always following the mixer, is described by a pure time delay, $t_i$. The complete model equations and parameter values can be found in Appendix B and Komulainen et al. [56]. Here, the SX process considers the following operating conditions: organic flow ($F_{LO}$) of 17.83 m$^3$/min, electrolyte flow ($F_{1e}$) of 6.26 m$^3$/min, aqueous flow ($F_{1a}$) of 16.88 m$^3$/min with concentration ($c_{oa}$) of 1.53 g/L, and aqueous flow ($F_{2a}$) of 16.88 m$^3$/min with concentration ($c_{2a}$) of 3.37 g/L. The SX process model was simulated using Simulink (see Figure 9b–f), obtaining $c_{1e}$, $c_{1a}$, $c_{3a}$, $c_{3o}$, and $c_{4o}$ values of 51.58, 0.254, 1.38, 6.62, 1.01 and 3.532 g/L, respectively.

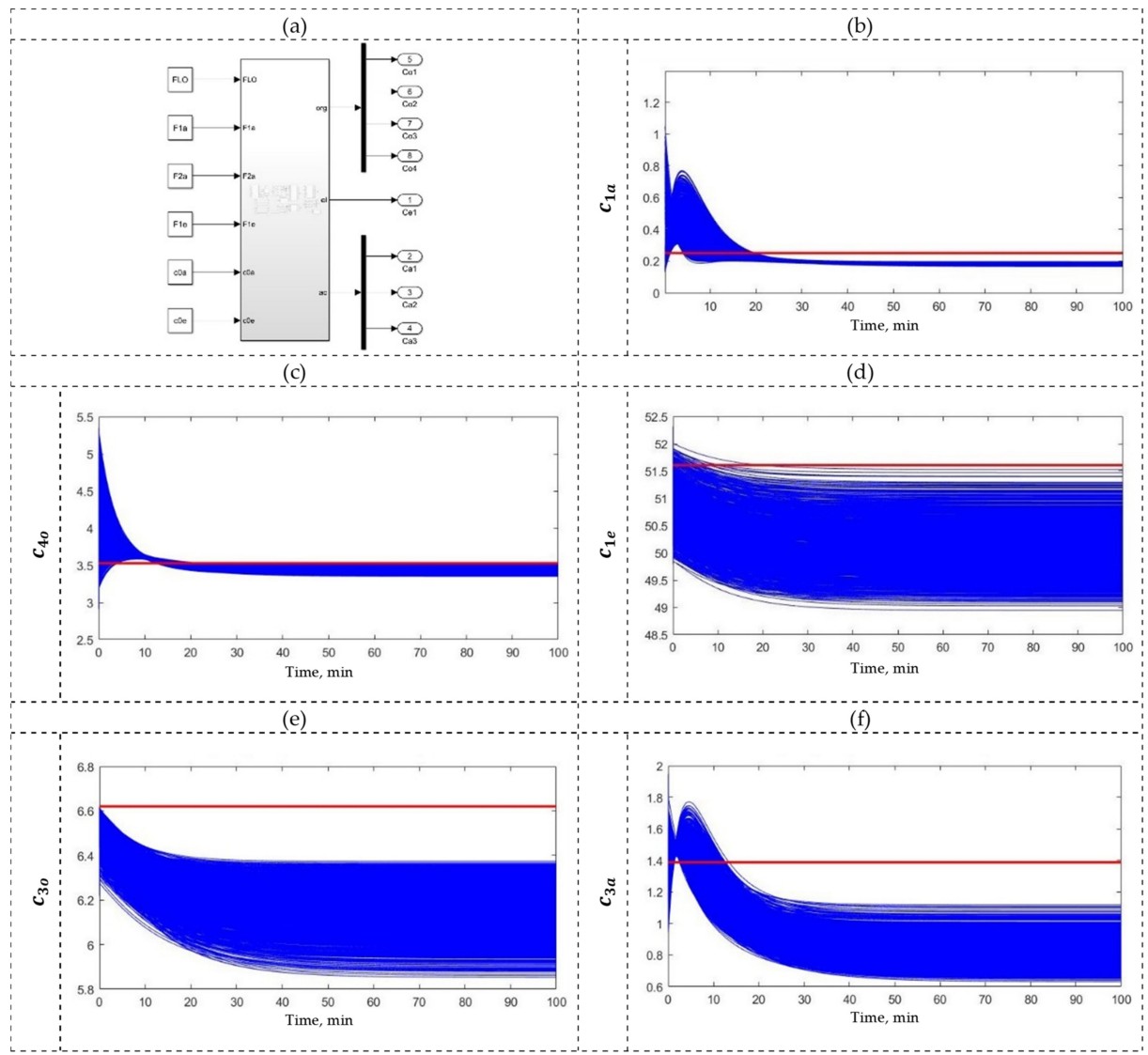

**Figure 9.** (**a**) SX plant as a MIMO system, and UA results using (**b**) $c_{1a}$, (**c**) $c_{4o}$, (**d**) $c_{1e}$, (**e**) $c_{3o}$, and (**f**) $c_{3a}$ as output variables.

Step 2. UA and GSA

For the MIMO system (see Figure 9a), the main output variables are $c_{1e}$, $c_{1a}$, $c_{3a}$, $c_{3o}$, and $c_{4o}$, and the possible manipulated input variables are $F_{LO}$, $F_{1a}$, $F_{2a}$, $F_{1e}$, $c_{0a1}$, $c_{0a2}$, and $c_{0e}$. The manipulated input variables of the SX process exhibit epistemic uncertainty due to insufficient information collected from the related literature. Again, according to the principle of indifference, a uniform distribution must be implemented to describe the epistemic uncertainty in the absence of information. In this way, manipulated input variables were described as follows: $FLO \sim U[16.04, 19.6]$ m$^3$/h, $F_{1a} \sim U[15.19, 18.56]$ m$^3$/h, $F_{2a} \sim U[14.19, 18.56]$ m$^3$/h, $F_{1e} \sim U[5.63, 6.88]$ m$^3$/h, $c_{0a1} \sim N[1.53, 0.5]$ %, and $c_{0a2} \sim N[3.37, 0.5]$%.

Figure 9b–f shows the SX plant responses when subjected to UA, considering a sample of 7000 instances of operation. Here, it can be observed that $c_{1a}$ and $c_{4o}$ present values similar to those obtained using nominal operating conditions despite the uncertainty, while $c_{1e}$, $c_{3o}$, and $c_{3a}$ exhibit lower values than the responses obtained using nominal operating conditions (red line), which indicates the influence of operating uncertainty on SX plant responses. The estimates provided by the SX model under uncertainty are consistent with

the related literature [24,56]. The next step is to study the uncertainty apportion using the Sobol–Jansen method.

Figure 10 shows the normalized total sensitivity indices of the input variables for each output variable of the SX plant. According to this figure, $F_{LO}$, $F_{1a}$, and $F_{1e}$ have an influence on $c_{1o}$, $c_{1a}$, and $c_{1e}$, respectively, and they should be selected as manipulated input variables, while $c_{0a1}$, $c_{0a2}$, and $c_{0e}$ should be selected as disturbances.

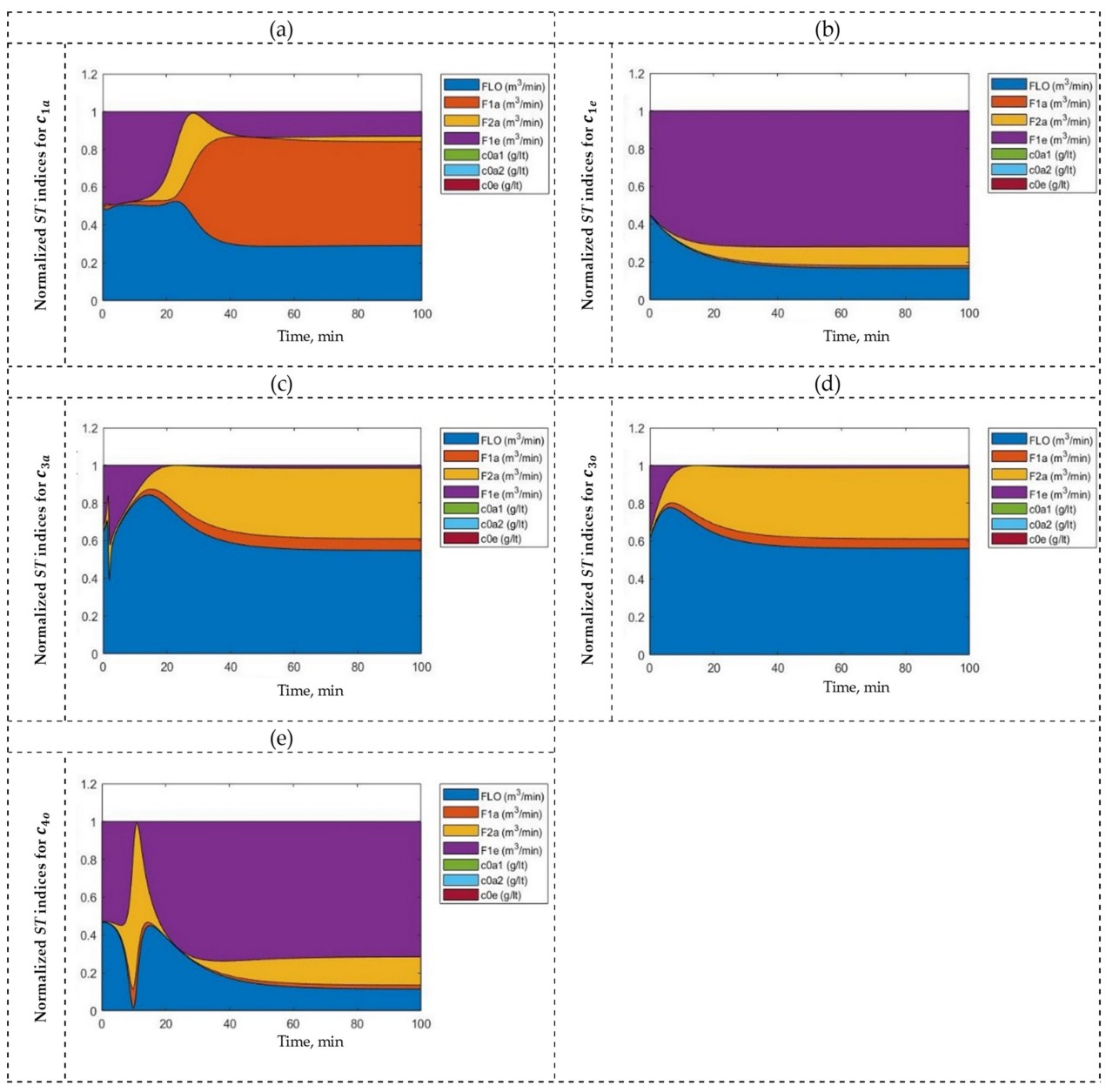

**Figure 10.** GSA using (**a**) $c_{1a}$, (**b**) $c_{1e}$, (**c**) $c_{3a}$, (**d**) $c_{3o}$, and (**e**) $c_{4o}$ as output variables.

Step 3. CSD

In this step, the variables of the process are classified as manipulated, controlled, supervised, or disturbed, allowing the SX model to be expressed in a standard control notation (Figure 3) as follows:

$$\dot{x} = f(x, u, d, p), \quad x(0) = x_0 ; \quad \dim(x) = 8, \quad \dim(u) = n_u$$
$$y = h(x) ; \dim(y) = n_y \tag{12}$$

where the vector-valued function of time, $f$, is defined by the right-hand side of Equation (11). For the SX process, the states ($x$) correspond to concentrations $c_{10}$, $c_{20}$, $c_{30}$, $c_{40}$, $c_{1a}$, $c_{2a}$, $c_{3a}$, and $c_{1e}$, while the manipulated input variables ($u$) are selected according to the output variables ($y$) to be regulated at desired values.

For the MIMO system of the SX process (see Figure 11a), and again for comparison purposes, the two structures are defined as follows:

(a)   A traditional structure (4 × 5) reported previously [24], with $n_u = 4$ and $n_y = 5$:

$$u = \begin{bmatrix} F_{LO} \\ F_{1a} \\ F_{2a} \\ F_{1e} \end{bmatrix}, \ y = \begin{bmatrix} c_{1e} \\ c_{1a} \\ c_{3a} \\ c_{3o} \\ c_{4o} \end{bmatrix}, \ d = \begin{bmatrix} c_{0a1} \\ c_{0a2} \\ c_{0e} \end{bmatrix} \tag{13}$$

(b)   A reduced structure (3 × 3) obtained in step 2, with $n_u = n_y = 3$:

$$u = \begin{bmatrix} F_{LO} \\ F_{1a} \\ F_{1e} \end{bmatrix}, \ y = \begin{bmatrix} c_{1e} \\ c_{1a} \\ c_{3o} \end{bmatrix}, \ d = \begin{bmatrix} c_{0a1} \\ c_{0a2} \\ c_{0e} \end{bmatrix} \tag{14}$$

These two CSDs are proposed using total sensitivity indices, and again, two control strategies are considered to evaluate performance and robustness: a conventional controller (PID) and an advanced controller (MPC). In the next step, three control systems for the SX plant are evaluated and analyzed, as shown in Figure 11: 3 × 3 PID, 3 × 3 MPC, and 4 × 5 MPC.

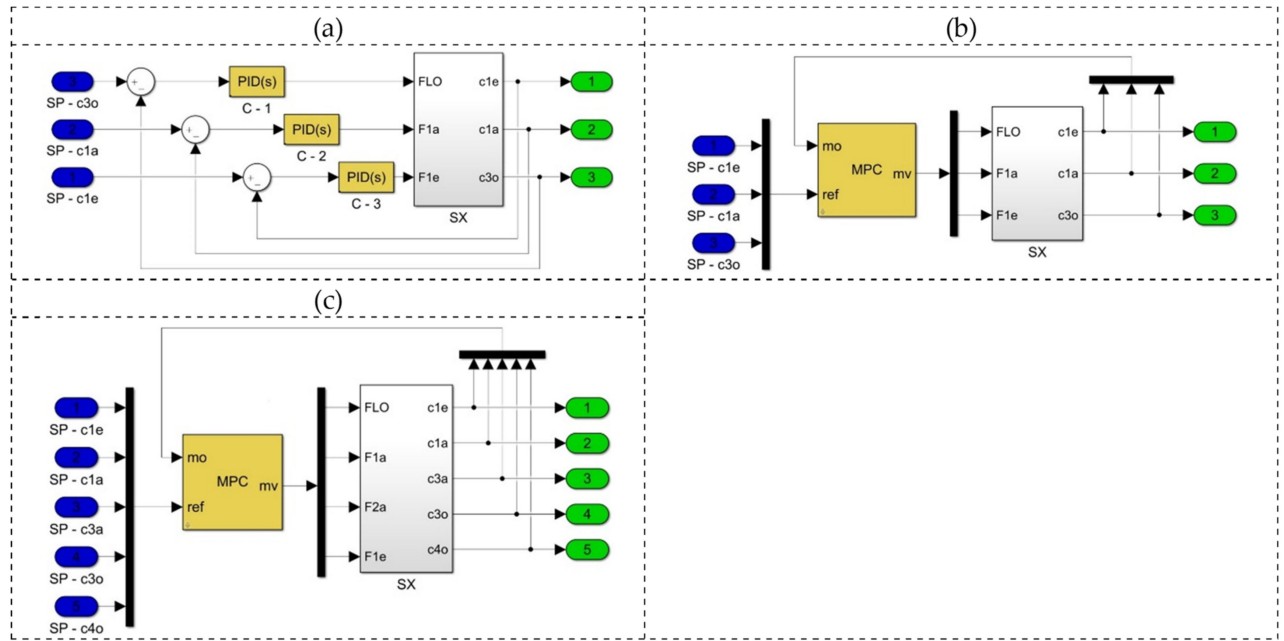

**Figure 11.** Control systems for SX plant: (**a**) 3 × 3 PID, (**b**) 3 × 3 MPC, (**c**) 4 × 5 MPC.

Step 4. Closed-loop validation

The designed control systems were implemented in MATLAB and compared with their corresponding open-loop dynamics, as shown in Figure 11. Again, the parameters of the PID and MPC controllers were determined via the automatic tuning tool included in Simulink. In Figure 12, it can be seen that the dynamic responses of all variables for the 3 × 3 PID and 3 × 3 MPC are satisfactory, while the performance of the 4 × 5 MPC

controller is poor. To quantify the quality of these results, IAE was calculated, and the values are given in Table 2.

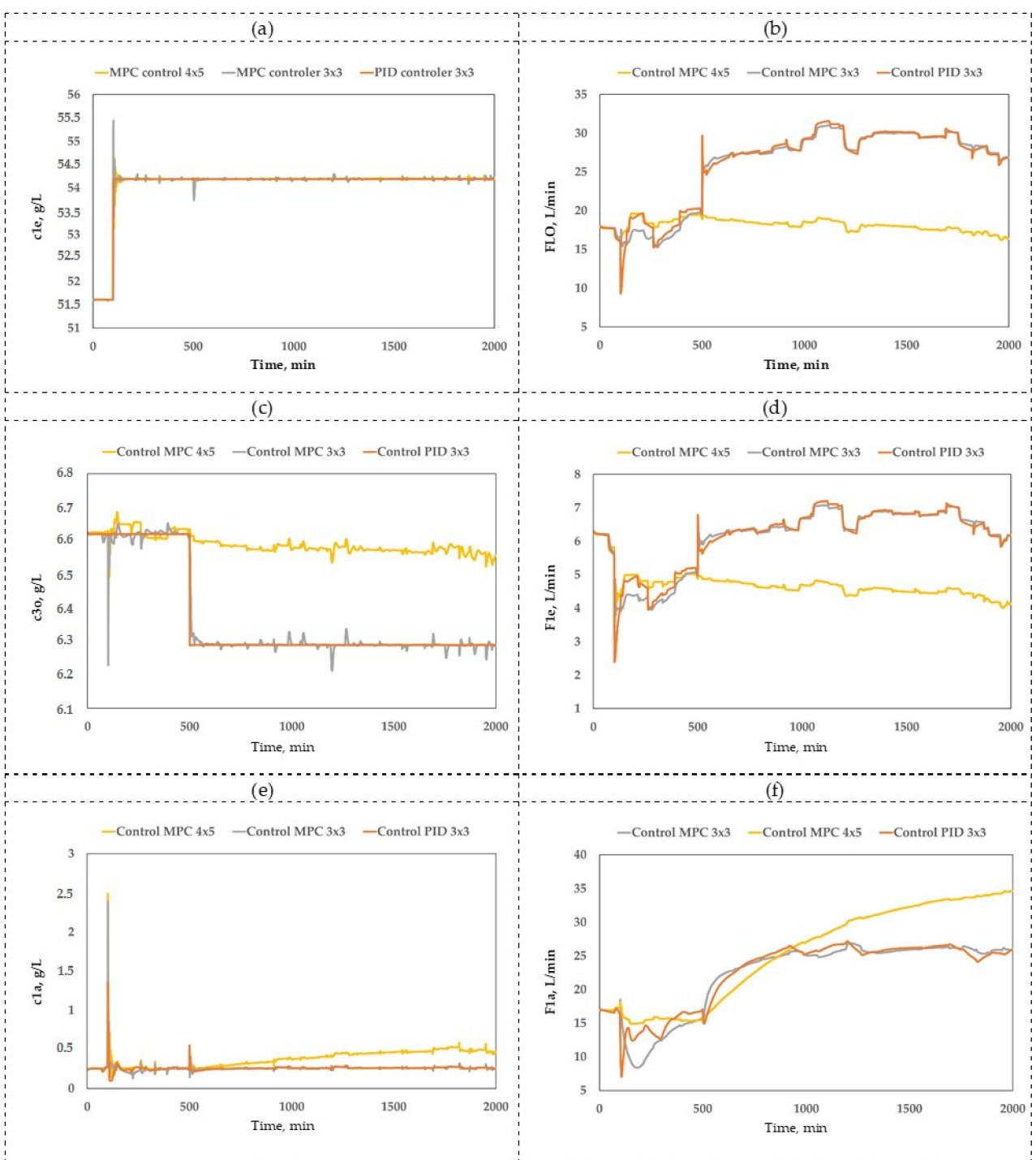

**Figure 12.** Dynamic response of controlled variables (**a**) $c_{1e}$, (**c**) $c_{3o}$, and (**e**) $c_{1a}$ and manipulated variables (**b**) $F_{LO}$, (**d**) $F_{1e}$, and (**f**) $F_{1a}$.

In the case of $c_{1e}$, Table 2 shows that the $3 \times 3$ PID, $3 \times 3$ MPC, and $4 \times 5$ MPC controllers provide IAE values of 0.008, 0.096, and 0.112, respectively, for a step change in the set point. In other words, the PID controller designed using total sensitivity indices exhibits high performance compared to the other controllers, which is related to the strong influence of $F_{1e}$ on $c_{1e}$. In the case of $c_{1a}$, Table 2 shows that the $3 \times 3$ PID, $3 \times 3$ MPC, and

$4 \times 5$ MPC controllers provide IAE values of 0.072, 0.090, and 2.234, respectively. Again, the controller designed using total sensitivity indices shows high performance compared to the other controllers, even though $FL_o$ and $F_{1e}$ have a side effect on $c_{1a}$. In the case of $c_{3o}$, Table 2 shows that the $3 \times 3$ PID, $3 \times 3$ MPC, and $4 \times 5$ MPC controllers provide IAE values of 0.001, 0.060, and 2.636, respectively. Here, the controller designed using total sensitivity indices exhibits high performance compared to MPC despite the secondary influence of $F_{2a}$ on $c_{3o}$. The good and sufficient performance of MPC could be explained by the cross-influence of some input variables on the SX plant responses detected in the GSA (Figure 10).

**Table 2.** Analysis and interpretation of results obtained by controllers, SX plant.

| Control | Controlled Variables | | | Manipulated Variables | | |
|---|---|---|---|---|---|---|
| IAE($10^{-3}$) | $c_{1e}$ | $c_{1a}$ | $c_{3o}$ | $F_{1e}$ | $F_{1a}$ | $FL_o$ |
| High performance | PID $3 \times 3$ 0.008 | PID $3 \times 3$ 0.072 | PID $3 \times 3$ 0.001 | MPC $3 \times 3$ 2.300 | MPC $3 \times 3$ 9.088 | MPC $3 \times 3$ 9.365 |
| Good performance | MPC $3 \times 3$ 0.096 | MPC $3 \times 3$ 0.090 | MPC $3 \times 3$ 0.060 | PID $3 \times 3$ 2.545 | PID $3 \times 3$ 10.230 | PID $3 \times 3$ 10.660 |
| Sufficient performance | MPC $4 \times 5$ 0.112 | MPC $4 \times 5$ 2.234 | MPC $4 \times 5$ 2.636 | MPC $4 \times 5$ 21.598 | MPC $4 \times 5$ 85.689 | MPC $4 \times 5$ 110.112 |
| Reference | 2.180 | Open-loop 0.099 | 0.605 | | | |

## 4. Conclusions

A methodology was presented to design decentralized control structures. This methodology considers the use of GSA based on the Sobol–Jansen method to establish the control structure design (input–output pairing) for MIMO systems operating under uncertainty conditions. These control structures are made using total sensitivity indices provided by the Sobol–Jasen method, and their behavior depends on the dynamics of the studied process and the magnitude of the uncertainty. In this sense, the Sobol–Jansen method provides graphical results that help in understanding the dynamic behavior of systems under uncertainty. The methodology was illustrated using a SAG mill and an SX plant operating under uncertainty. For the SAG mill, the methodology allowed us to design a $2 \times 2$ decentralized control structure whose pairings J/F and $M_p/\phi_c$ exhibited high and sufficient performance, respectively, compared to MPC. For the SX plant, the methodology allowed us to design a $3 \times 3$ decentralized control structure whose pairings $c_{1o}/F_{LO}$, $c_{1a}/F_{1a}$, and $F_{1e}/c_{1e}$ exhibited high performance compared to MPC. The proposed methodology for the design of the control structure using GSA was illustrated with mineral processes, and it can be applied to any other process that operates under uncertainty; therefore, it could provide satisfactory results for a wide range of operating conditions.

**Author Contributions:** Conceptualization, F.A.L. and L.A.C.; methodology, F.A.L. and O.M.-Q.; software, F.A.L. and O.M.-Q.; formal analysis, F.A.L., O.M.-Q., L.A.C. and T.L.-A.; investigation, O.M.-Q. and F.A.L.; data curation, O.M.-Q. and F.A.L.; writing—Original draft preparation, F.A.L. and O.M.-Q.; writing—Review and editing, O.M.-Q., F.A.L., L.A.C. and T.L.-A.; funding acquisition, L.A.C. All authors have read and agreed to the published version of the manuscript.

**Funding:** The authors are grateful for the support of Agencia Nacional de Investigación y Desarrollo de Chile (ANID) through Anillo-Grant No. ACT210027 and Fondecyt 1211498.

**Data Availability Statement:** Not applicable.

**Acknowledgments:** This publication was supported by ANID, Anillo-Grant ACT210027, and Fondecyt 1211498.

**Conflicts of Interest:** The authors declare no conflict of interest.

## Nomenclature

For SAG model:

| | |
|---|---|
| $A$ | ore impact breakage parameter |
| $a$ | parameter of specific breakage rate model |
| $a_S$ | parameter of specific breakage rate model |
| $B_{ij}$ | cumulative breakage distribution function |
| $B$ | values of input variables that provide desired behavior of output milling model |
| $B^-$ | values of input variables that provide unwanted behavior of output milling model |
| $b_{ij}$ | ore impact breakage parameter |
| $n$ | breakage distribution function |
| $C^*$ | number of species present in fresh feed |
| $c_i$ | mass flow recirculated internally by grill |
| $c_f$ | classification efficiency of internal grid mill |
| $c_f$ | solid weight percentage in mill charge |
| $D$ | mill diameter |
| $E_{cs}$ | comminution specific energy (kWh/t) |
| $F$ | fresh ore flux fed to mill, t/h |
| $f_i$ | fraction of fresh ore flux fed to mill |
| $J$ | fraction of mill filling |
| $J_b$ | percentage of mill volume occupied by steel balls |
| $K_i$ | specific breakage rate |
| $L$ | mill length |
| $M$ | parameter of classification efficiency model |
| $M_p$ | mill power consumption |
| $m$ | total number of input variables in model $Y$ |
| $R$ | total number of simulations |
| $S_j$ | first-order sensitivity index for input variable $x_j$ |
| $S_j^T$ | total sensitivity index for input variable $x_j$ |
| $V$ | mill volume |
| $R$ | total number of simulations |
| $V(Y)$ | variance of model $Y$ |
| $W$ | mass retained in mill |
| $W_a$ | water in mill charge |
| $w_a$ | ratio between ore mass and water mass retained inside mill |
| $w_i$ | weight fraction of retained mass in mill |
| $Y_d$ | percentage of solids in discharge mill |
| $x_0$ | parameter of specific breakage rate model |
| $x_i$ | particle size of species present in fresh feed |
| $x_{50}$ | parameter of classification efficiency model |
| $Z$ | parameter of classification efficiency model |
| $\alpha$ | characteristic parameter of material |
| $\alpha_1$ | parameter of specific breakage rate model |
| $\alpha_s$ | parameter of specific breakage rate model |
| $\beta$ | fraction of fines produced in a single fracture event |
| $\beta_1$ | parameter of classification efficiency model |
| $\gamma$ | parameter of cumulative breakage distribution function |
| $\mu$ | parameter of specific breakage rate model |
| $\Psi$ | parameter of specific breakage rate model |
| $\Lambda$ | parameter of cumulative breakage distribution function |
| $\varnothing_j$ | percentage of critical speed |
| $\varnothing_c$ | parameter of classification efficiency model |

For SX model:

| | |
|---|---|
| $F_{1a}$ | flow inputs to process, pregnant leach solution (PLS) |
| $F_{2a}$ | flow inputs to process, pregnant leach solutions (PLS) |
| $c_{0a}$ | solution with a copper concentration |
| $F_{1e}$ | poor electrolyte flow |

## Appendix A. SAG Model

The mathematical expression used to model the breakage distribution function $b_{ij}$ is based on $B_{ij}$, the cumulative breakage distribution function, given by:

$$B_{ij} = \phi_j \left( \frac{x_{i-1}}{x_j} \right)^{\gamma} + (1 - \phi_j) \left( \frac{x_{i-1}}{x_j} \right)^{\beta} \tag{A1}$$

where $\phi_j$ and $\gamma$ are parameters with values ranging from 2.5 to 5 and 0.5 to 1.5, respectively, and $\beta$ represents the fraction of fines produced in a single fracture event. Considering that $B_{ij}$ is a cumulative distribution, this can be expressed as:

$$B_{i,j} = b_{n,j} + b_{n-1,j} + \ldots + b_{i,j} = \sum_{k=i}^{n} b_{kj} \tag{A2}$$

so that

$$B_{i,j} - B_{i+1,j} = b_{ij} \tag{A3}$$

The classification efficiency of the internal grid mill, $c_i$, is calculated with the following expression:

$$c_i = \psi \beta (x_i M)^{(\beta-1)} exp\left( -\psi (x_i M)^{\beta} \right) + \frac{1}{1 + \left( \frac{x_{50}}{x_i} \right)^{Z}} \tag{A4}$$

where $\psi$, $Z$, $M$, $x_{50}$, and $\beta$ are parameters of the model. In the case of specific breakage rate $K_i$, related theory suggests that this parameter varies with particle size; the typical form of specific breakage rate has three regions. Magne et al. [48,49] proposed the following equation to estimate the specific breakage rate in the three regions:

$$K_i = a \left( \frac{x_i}{x_0} \right)^{\alpha} \frac{1}{1 + \left( \frac{x_i}{\mu} \right)^{\Lambda}} + a_s \left( \frac{x_i}{x_0} \right)^{\alpha_s} \tag{A5}$$

where $\alpha$, $\mu$, $\Lambda$, $a$, $a_s$, $x_0$, and $\alpha_s$ are parameters of the model.

## Appendix B. SX Model

The mass balances of the SX process are given below.
Organic–aqueous balance in E1P:

$$\frac{dc_1^{org}(t)}{dt} = \frac{F_1^{org}(t)}{V_{mix,1}(t)} \cdot \left[ c_4^{org}(t - t_0) - c_1^{org}(t) \right] + K_1 \left[ c_1^{org}(t) - c_1^{org^*}(t) \right] \tag{A6}$$

$$\frac{dc_1^{aq}(t)}{dt} = \frac{F_1^{aq}(t)}{V_{mix,1}(t)} \cdot \left[ c_0^{aq}(t) - c_1^{aq}(t) \right] - K_1 \left[ c_1^{org}(t) - c_1^{org^*}(t) \right] \tag{A7}$$

Organic–aqueous balance in E1S:

$$\frac{dc_2^{org}(t)}{dt} = \frac{F_2^{org}(t)}{V_{mix,2}(t)} \cdot \left[ c_1^{org}(t - t_1) - c_2^{org}(t) \right] + K_2 \left[ c_2^{org}(t) - c_2^{org^*}(t) \right] \tag{A8}$$

$$\frac{dc_3^{aq}(t)}{dt} = \frac{F_2^{aq}(t)}{V_{mix,2}(t)} \cdot \left[ c_2^{aq}(t) - c_3^{aq}(t) \right] - K_2 \left[ c_2^{org}(t) - c_2^{org^*}(t) \right] \tag{A9}$$

Organic–aqueous balance in E2S:

$$\frac{dc_3^{org}(t)}{dt} = \frac{F_3^{org}(t)}{V_{mix,3}(t)} \cdot \left[ c_2^{org}(t - t_2) - c_3^{org}(t) \right] + K_3 \left[ c_3^{org}(t) - c_3^{org^*}(t) \right] \tag{A10}$$

$$\frac{dc_2^{aq}(t)}{dt} = \frac{F_2^{aq}(t)}{V_{mix,3}(t)} \cdot \left[c_0^{aq}(t) - c_2^{aq}(t)\right] - K_3\left[c_3^{org}(t) - c_3^{org^*}(t)\right] \tag{A11}$$

Organic–electrolyte balance in S1H:

$$\frac{dc_4^{org}(t)}{dt} = \frac{F_4^{org}(t)}{V_{mix,4}(t)} \cdot \left[c_3^{org}(t - t_3) - c_4^{org}(t)\right] - K_4\left[c_1^{el}(t) - c_1^{el^*}(t)\right]$$
$$\frac{dc_1^{el}(t)}{dt} = \frac{F_1^{el}(t)}{V_{mix,4}(t)} \cdot \left[c_0^{el}(t) - c_1^{el}(t)\right] + K_4\left[c_1^{el}(t) - c_1^{el^*}(t)\right] \tag{A12}$$

From the described balance equations, the following relationships are established:

$$FLO = F_1^{org} = F_2^{org} = F_3^{org} = F_4^{org} \tag{A13}$$

$$c(RE) = c_1^{el}(t - t_4) \tag{A14}$$

$$c(RaffP) = c_1^{aq}(t - t_1) \tag{A15}$$

$$c(RaffS) = c_3^{aq}(t - t_2) \tag{A16}$$

$$c(LO) = c_3^{org}(t - t_3) \tag{A17}$$

$$c(BO) = c_4^{org}(t - t_4) \tag{A18}$$

Theoretical equilibrium values for extraction and re-extraction are determined from the McCabe–Thiele diagram, presented in Figure A1. Isotherms assume that extraction and re-extraction are constant, while the operating lines are constantly changing according to the organic/aqueous volume ratio in each tank. The equilibrium value in a tank is the point of coincidence of the equilibrium isotherm and the inverse operating line, weighted by the efficiency coefficient $\alpha$.

In extraction, the equilibrium isotherm is not linear:

$$c^{org} = Ac^{aq}/(B + c^{aq}) \tag{A19}$$

and in re-extraction it is linear:

$$c^{org} = C \cdot c^{aq} + D \tag{A20}$$

In extraction, the theoretical equilibrium point (100% efficiency) is determined by:

$$y = 1/2a\left(-(Ba - A + b) - \sqrt{(Ba - a + b)^2 - 4aBb}\right) \tag{A21}$$

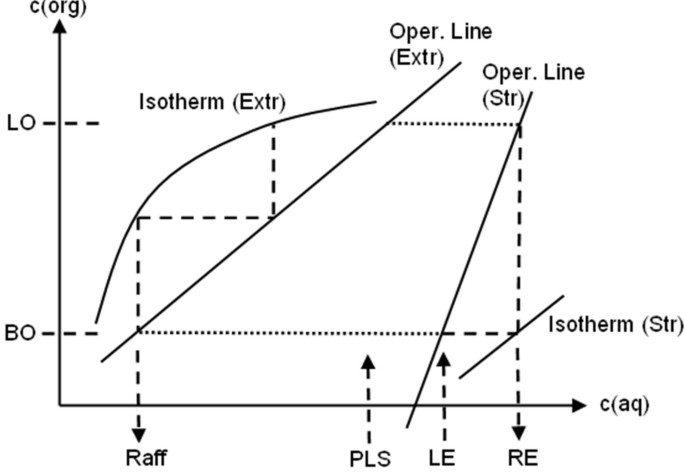

**Figure A1.** McCabe–Thiele diagram: two extraction stages and one re-extraction stage (adapted from [56]).

The equilibrium value for aqueous and organic concentrations is the weighted efficiency of the theoretical value:

$$c^{aq*} = \alpha_i x^* + (1 - \alpha_i) \cdot c_0^{aq} \tag{A22}$$

$$c^{org*} = \alpha y^* + (1 - \alpha) \cdot c_0^{org} = \alpha \cdot (ax^* + b) + (1 - \alpha) \cdot c_0^{org} \tag{A23}$$

Here, *a* and *b* are parameters of the equilibrium isotherm, where the slope *a* of the operating line is:

$$a = -F^{aq} / F^{org} \tag{A24}$$

while *b* is the linear term that combines the input concentrations of the organic and aqueous phases:

$$b = c_0^{org} - a \cdot c_0^{aq} \tag{A25}$$

In re-extraction, the isotherm parameters are *C* and *D* and *a* and *b*; in the same way, the equilibrium point is solved by:

$$y = Cx + D = ax + b \tag{A26}$$

resulting in the theoretical equilibrium concentration of the aqueous phase $x^*$:

$$x^* = (b - D) / (C - a) \tag{A27}$$

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
