# Peer review of "Control Structure Design Using Global Sensitivity Analysis for Mineral Processes under Uncertainties"

_minerals, doi:10.3390/min12060736_

Round 1

Reviewer 1 Report

1.    The text must be proofread in case of language.
2.    Referencing format must be consistent throughout the text and based on the instruction of the journal.
3.    Probably the time could be modified as “Control structure design using global sensitivity analysis for mineral processes under uncertainties”.
4.    A brief description of findings must be written in the abstract.
5.    Matrix ?? is created in the Sobol-Jansen method but is not involved in Equations 4 and 5. A clarification is required in the manuscript.
6.    The term classification efficiency (ci) is not involved in Equation 6. 
7.    In the manuscript, line 295, Figure 5 must be corrected as Figure 4.
8.    Please recheck the Figure 4 in line 296.
9.    The necessity of covering two case studies must be described in the text. (Probably to be able to construct a matrix with higher degrees and show the capability of the model!).
10.    Line 427, C1o must be mixed up with C4o. Please recheck it.
11.    Although the application of other methods and references in this research is clear, the novelty of the research is not clearly described. This must be strengthened and highlighted in the manuscript. (Probably Step 4: closed-loop validation needs to be discussed and clarified more!).

Author Response

1 The text must be proofread in case of language.

R. The work was sent to English editing.

2. Referencing format must be consistent throughout the text and based on the instruction of the journal.

R. The references were improved throughout the manuscript.

3.   Probably the time could be modified as “Control structure design using global sensitivity analysis for mineral processes under uncertainties”.

R. The authors accessed the suggestion, then the title of the manuscript was modified.

4.  A brief description of findings must be written in the abstract.

R. The following comment was added in the abstract “The results indicate that the methodology allows the design of 2 × 2 and 3 × 3 decentralized control structures for the SAG mill and SX plant, respectively, which exhibit good performance compared to MPC. For example, for the SAG mill, the determined pairings were fresh ore flux/fraction of mill filling and power consumption/percentage of critical speed.”

5. Matrix ?? is created in the Sobol-Jansen method but is not involved in. Equations 4 and 5. A clarification is required in the manuscript.

R. The literature proposes several methods to estimate the sensitivity indices; some implement one or another matrices A, B, C, and D defined in the manuscript; for example, the Sobol-Jansen method implements A, B, and D matrix.

6. The term classification efficiency (ci) is not involved in Equation 6. 

R. The following comment was made in the manuscript “is the classification efficiency of the internal grid, which affects the mass flow recirculated internally.”

7. In the manuscript, line 295, Figure 5 must be corrected as Figure 4.

R. The Figure cited was checked.

8.  Please recheck the Figure 4 in line 296.

R. The Figure cited was checked.

9. The necessity of covering two case studies must be described in the text. (Probably to be able to construct a matrix with higher degrees and show the capability of the model!).

R. The following text was added in the manuscript “Two instances were studied to demonstrate the methodology’s capacity to address systems with different degrees of freedom.”

10. Line 427, C1o must be mixed up with C4o. Please recheck it.

R. The nomenclature was checked

11. Although the application of other methods and references in this research is clear, the novelty of the research is not clearly described. This must be strengthened and highlighted in the manuscript. (Probably Step 4: closed-loop validation needs to be discussed and clarified more!).

R. The following sentence was added throughout the manuscript “GSA has not been applied for decentralized control structures designing yet; GSA allows us to understand the dynamic behavior of systems under uncertainty in broad values range, unlike approaches proposed in the literature.”

Reviewer 2 Report

  1. The focus of the abstract is misplaced. The origin of some research methods takes up too much space, so that there is basically no summary in results of the last two case application.
  2. The lack of specific introduction to the data sources of the two cases may easily make readers doubt the authenticity of the research content.
  3. How the first-order dynamics on line 257 is determined, as far as I know, there is not only first-order kinetic in the grinding process.
  4. For the validation of this method, it is commonly necessary to provide actual data for comparison with simulated data, but I did not find these in this paper.
  5. The appendix A attached at the end has no effect on the understanding of this paper, and I did not notice which part applies these formulas, please explain it.

Author Response

1. The focus of the abstract is misplaced. The origin of some research methods takes up too much space, so that there is basically no summary in results of the last two case application.

R. The following sentence was added in the abstract “Results indicate that the methodology allows designing 2x2 and 3x3 decentralized control structures for the SAG mill and SX plant, respectively, which exhibit a high performance compared to MPC; for example, for the SAG mill, the pairing determined were fresh ore flux/fraction of mill filling and power consumption/percentage of critical speed.”

2. The lack of specific introduction to the data sources of the two cases may easily make readers doubt the authenticity of the research content.

R. The data sources were [1,2] and [3,4] for the SAG mill and SX plant, respectively. Consequently, models implemented in the manuscript were tuned aiming to emulate cited works. It is worth mentioning that the SAG mill’s mathematical model was validated in the manuscript [5], where the authors reported instances similar to those used in the current work.

3. How the first-order dynamics on line 257 is determined, as far as I know, there is not only first-order kinetic in the grinding process.

R. You are right; some authors indicate that second-order kinetic could model the breakage of coarse particles better than first-order kinetic [6,7]. However, computational experiments were carried out under conditions where first-order kinetic provides reliable estimations [1,2].

4. For the validation of this method, it is commonly necessary to provide actual data for comparison with simulated data, but I did not find these in this paper.

R. We have no operating dataset to validate the designed structures, which is not unusual in control works based on simulation systems; in fact, this is observed in the following manuscripts [8–12]. Despite this disadvantage, the designed control structures via the methodology were compared with MPC controller, which is widely accepted by the process control community due to its robustness.

5. The appendix A attached at the end has no effect on the understanding of this paper, and I did not notice which part applies these formulas, please explain it.

R. The following text was added in the manuscript “The complete model equations and parameter values can be found in Appendix A and Lucay et al. [13]. In the appendix, the reader can see the expressions used to model the cumulative breakage distribution function and its implementation to determine , as well as expressions used to estimate the classification efficiency of the internal grid mill, which is required to calculate recirculated mass flow. In [30], the reader can find comminution-specific energy, mill power consumption, and the fraction of mill filling expressions, among other equations.”
